# TRPA1 Channel is Involved in SLIGRL-Evoked Thermal and Mechanical Hyperalgesia in Mice

**DOI:** 10.3390/medsci7040062

**Published:** 2019-04-18

**Authors:** Merab G. Tsagareli, Ivliane Nozadze, Nana Tsiklauri, Gulnaz Gurtskaia

**Affiliations:** Laboratory of Pain and Analgesia, Beritashvili Center for Experimental Biomedicine, Tbilisi 0160, Georgia; nozadzelia@yahoo.com (I.N.); nanatsiklauri1812@gmail.com (N.T.); gurtskaiaguliko@yahoo.com (G.G.)

**Keywords:** antinociception, hyperalgesia, thermal withdrawal, mechanical withdrawal, pruritus, SLIGRL, TRPA1

## Abstract

Persistent itch (pruritus) accompanying dermatologic and systemic diseases can significantly impair the quality of life. It is well known that itch is broadly categorized as histaminergic (sensitive to antihistamine medications) or non-histaminergic. Sensory neurons expressing Mas-related G-protein-coupled receptors (Mrgprs) mediate histamine-independent itch. These receptors have been shown to bind selective pruritogens in the periphery and mediate non-histaminergic itch. For example, mouse MrgprA3 responds to chloroquine (an anti-malarial drug), and are responsible for relaying chloroquine-induced scratching in mice. Mouse MrgprC11 responds to a different subset of pruritogens including bovine adrenal medulla peptide (BAM8–22) and the peptide Ser-Leu-Ile-Gly-Arg-Leu (SLIGRL). On the other hand, the possibility that itch mediators also influence pain is supported by recent findings that most non-histaminergic itch mediators require the transient receptor potential ankyrin 1 (TRPA1) channel. We have recently found a significant increase of thermal and mechanical hyperalgesia induced by non-histaminergic pruritogens chloroquine and BAM8–22, injected into mice hindpaw, for the first 30–45 min. Pretreatment with TRPA1 channel antagonist HC-030031 did significantly reduce the magnitude of this hyperalgesia, as well as significantly shortened the time-course of hyperalgesia induced by chloroquine and BAM8–22. Here, we report that MrgprC11-mediated itch by their agonist SLIGRL is accompanied by heat and mechanical hyperalgesia via the TRPA1 channel. We measured nociceptive thermal paw withdrawal latencies and mechanical thresholds bilaterally in mice at various time points following intra-plantar injection of SLIGRL producing hyperalgesia. When pretreated with the TRPA1 antagonist HC-030031, we found a significant reduction of thermal and mechanical hyperalgesia.

## 1. Introduction

Itch is an unpleasant skin sensation that evokes the desire to scratch. Among different pain reactions that lead to avoidance of noxious stimuli, itch is primarily thought to be a means for eliminating exogenous compounds such as parasites and plant particles. Itch sensation and scratching behaviors are conserved across a broad range of species, from rodents and birds to humans. In the latter, environmental substances such as allergens, mosquito bites, and some chemical compounds can cause itch, but chronic itch can accompany systemic diseases including atopic dermatitis (AD), kidney failure, cholestasis, and neuronal lesions, and can significantly impair the quality of life [1,2].

G-protein-coupled receptors (GPCRs) are the major class of sensory proteins working as complex intracellular signaling messengers. An understanding of the dynamic nature of GPCRs within primary sensory neurons and neighboring cells brings new insights into their contributions to the physiology and pathophysiology of pain and itch and provides novel opportunities for therapeutic intervention [3,4].

Protease-activated receptors (PARs) are activated by protease-induced cleavage of a part of the extracellular domain that acts as a tethered ligand and their role in pain and itch has received considerable attention. Cowhage spicules from the bean plant (Mucuna pruriens) have long been known to induce itch. Its active component is mucunain, a cysteine protease, which acts at PAR-2 and PAR-4 subtypes to produce histamine-independent itch [3,4,5]. A tethered ligand, such as the peptide Ser-Leu-Ile-Gly-Arg-Leu (SLIGRL) (agonist of PAR-2) is also known to elicit scratching in mice [3,4,5,6]. Recent studies reported that SLIGRL elicited scratching via Mas-related G-protein-coupled receptors C11 (MrgprC11) rather than PAR-2 [3,4,5,6,7]. Chloroquine and the bovine adrenal medulla peptide 8–22 (BAM8–22) produce itch-related scratching through MrgprA3 and MrgprC11, respectively, in mice and in humans [3,4,5]. In comparison with responses of control mice, contact hypersensitivity mice exhibited a significant increase in the scratching evoked by BAM8–22, a peptide that elicits a histamine-independent itch [3,4,5,6,7,8].

Numerous experiments have been conducted to investigate the role of transient receptor potential (TRP) channels in sensory signal transductions. These studies presented evidence showing that the peripheral nervous system in particular transduces cutaneous sensory stimuli into electrical signals and transmits them to the central nervous system (CNS). In general, mammalian TRP channels comprise six related protein subfamilies TRP-canonical (TRPC), TRP-vanilloid (TRPV), TRP-melastatin (TRPM), TRP-ankyrin (TRPA), TRP-mucolipin (TRPML), and TRP-polycystic (TRPP). Although the physiological functions of most TRP channels are not well known, their wide distribution indicates that the biological functions and activation mechanisms for these channels are diverse. For instance, TRP channels are best recognized for their contributions to sensory transduction, response to temperature, nociceptive stimuli, touch, osmolarity, pheromones or odorants, and other stimuli from both within and outside the cell [9,10,11,12,13,14].

For the last decade, it was established that some TRP channels, especially the so-called thermo-sensitive TRP channels are involved in the detection of itch producing stimuli [1,2,4,15,16,17,18,19]. The possibility that itch mediators also influence pain is supported by recent findings that most non-histaminergic itch mediators require TRP ankyrin 1 (TRPA1) channel [20,21]. It is unclear whether most pruritogenic stimuli act through Mrgpr receptors, at least indirectly, but they have helped clarify the neural mechanisms underlying itch, and to a lesser extent, pain sensation.

The anti-malarial drug chloroquine produces itching as a common side effect. MrgprA3 is specific for chloroquine, and Mrgpr-cluster deletion mice showed normal nociception and histaminergic pruriception but did not exhibit chloroquine-induced itch. The BAM8–22 peptide also acts as a pruritogen and a selective agonist for MrgprC11. In studies of calcium signaling induced by chloroquine and BAM8–22, chloroquine-induced calcium increases in dorsal root ganglion (DRG) neurons were suggested to be caused by TRP channel activation. On the other hand, chloroquine- or BAM-induced intracellular calcium increases in HEK293 cells that heterologously express MrgprA3 or MrgprC11, respectively, were found to be due to calcium influx [22,23,24]. This identified, non-histaminergic mechanism of itch, including chloroquine- and BAM-induced calcium signaling, was investigated using TRPV1 knockout (KO) and TRPA1 KO mice and antagonists for TRPV1 and TRPA1. These experiments indicated that for chloroquine- and BAM-induced scratching behaviors in wild-type (WT) and TRPV1 KO mice, but not in TRPA1 KO mice, TRPA1-expressing neurons are required for both non-histaminergic itches [20]. Other experiments showed that chloroquine- and SLIGRL-induced itching is mediated through neurokinin-1 (NK-1) receptors by primary sensory neurons that are different from the neurons that transmit the itch sensation evoked by histamine or serotonin [21].

Recently, we have found that pretreatment with TRPA1 channel antagonist HC-030031 did significantly reduce hyperalgesia evoked by chloroquine and BAM8–22, injected into mice hindpaw [25]. In the present paper, we report that MrgprC11-mediated itch by its agonist SLIGRL is accompanied by heat and mechanical hyperalgesia via the TRPA1 channel.

## 2. Materials and Methods

### 2.1. Animals

The experiments were performed on wild-type male mice < 50 g in body weight, bred at the vivarium of Beritashvili Experimental Biomedicine Center. The animals were kept under standard housing conditions (22 ± 2 °C, 65% humidity, lights from 6:00 a.m. to 8:00 p.m.), and fed by standard dry diet; water freely available. Guidelines of International Association for the Study of Pain regarding investigations of experimental pain in conscious animal were followed throughout. The experimental protocol was approved by the local bioethics committee of the Beritashvili Experimental BMC. Experimental procedures were performed as described previously [25].

### 2.2. Chemical Injections

The peptide SLIGRL and the TRPA1 antagonist HC-030031 were purchased from Sigma-Aldrich Chemicals, Co. (St. Louis, MO, USA). SLIGRL was dissolved in saline to obtain 7.5, 15, and 40 mM/1 μL final dose, and HC-030031 was dissolved in 30 μL 1% DMSO and saline to obtain final dose 50 and 100 μg/30 μL. Various doses of this chemical were injected intra-plantar through a 30G needle connected by PE 50 tubing to a Hamilton micro-syringe. The same volumes of vehicle (isotonic saline) were microinjected in the same manner separately as a control. In the second set of experiments, 15–20 min prior to the start of the experiment, the same volume of the TRPA1 antagonist HC-030031 was pre-injected in the same hindpaw and animals were examined by the thermal and mechanical paw tests. Different animal groups were used for the experiments and they were tested with one concentration of irritant chemicals, antagonists, or vehicle, and not repeatedly used. Six mice were used for each group.

### 2.3. Thermal Paw Withdrawal (Hargreaves) Test

Mice first were habituated to stand on a glass surface heated to 30 °C within a Plexiglass enclosure, over three separate daily sessions. For formal testing, baseline latencies for paw withdrawals evoked by radiant thermal stimulation of each hind paw were measured minimum three times/paw, with at least 5 min elapsing between tests on a given paw. A light beam (Plantar Test 390, IITC, Woodland Hills, CA, USA) was focused onto the plantar surface of one hind paw through the glass plate from below, and the latency from onset of the light to brisk withdrawal of the stimulated paw was measured. The other hind paw was similarly tested 30–60 s later. The mouse was then held gently and one hind paw received an intra-plantar injection of chemicals or vehicle. The mice then were placed back onto the glass plate and withdrawal latencies of both paws were measured at 3, 15, 30, 45, 60, and 120 min post-injection of irritants.

### 2.4. Mechanical Paw Withdrawal Threshold (von Frey) Test

Mice were first habituated to standing on the mesh stand surface. For formal testing, baseline withdrawals were assessed using an Electronic von Frey Anesthesiometer (2390, IITC, CA, USA) filament that was pressed against the ventral paw from below. This device samples and holds force (g) at the moment that the hind paw was withdrawn away from the filament. Each paw was tested for baseline mechanical withdrawals at least three times, with at least 5 min elapsing between successive measurements of a given paw. The mouse then received a unilateral intra-plantar injection (see above) and was placed back onto the mesh stand surface. Mechanical paw withdrawals were measured at the same post-injection times as above for thermal paw withdrawals. The same groups of mice were used for thermal and mechanical withdrawal tests, with a minimum of 7 days in between successive tests to avoid possible carryover effects of stimuli.

### 2.5. Statistical Analysis

All data from behavioral tests were subjected to repeated measures of analysis of variance (ANOVA) and then were compared between chemicals and vehicle treatment groups by paired *t*-test. The data are expressed as mean ± S.E.M. Thereafter, Kruskal–Wallis ANOVA and subsequent Tukey test was used to assess differences between treatments. Statistical significance is acknowledged if *p* < 0.05. The statistical software utilized was InStat 3.05 (GraphPad Software, Inc, San Diego, CA, USA).

## 3. Results

Each of the 8 groups of mice received intra-plantar injection of saline in one hindpaw to establish baseline responses. Three days later, 3 groups of mice were injected with SLIGRL in the same hindpaw showing strong thermal and mechanical hyperalgesia that persisted beyond almost 2 h. These hyperalgesic effects are significant compared to the saline control group (*p* < 0.001) (Figure 1A,C). Four other groups of mice prior to injection of SLIGRL were pretreated with the TRPA1 channel antagonist HC-030031 (50 and 100 µg/30 µL). The unpaired two-tailed *t*-test showed that the difference of the mean value between TRPA1 antagonist pre-injected groups and only SLIGRL-injected groups at 5 min of post-SLIGRL injection is significant for thermal test, t = 4.228, df = 70, *p* < 0.0001, and more for mechanical test, t = 7.451, df = 70, *p* < 0.0001, respectively (Figure 2A,C). There was a significant difference in the extent of inhibition between 50 mg and 100 mg HC-030031 for 15 mM SLIGRL at 30–60 min of the post-SLIGRL injection (*p* < 0.001), but not for 40 mM SLIGRL (*p* > 0.05) in the thermal test (Figure 2A). These findings showed a significant attenuation of thermal and mechanical hyperalgesia for the first 30–45 min. There were mirror image hyperalgesic effects on the contralateral paw following SLIGRL injections in both the thermal and mechanical paw withdrawal tests (Figure 1B,D and Figure 2B,D).

## 4. Discussion

The obtained data showed a significant attenuation of thermal and mechanical hyperalgesia induced by the non-histaminergic pruritogen SLIGRL for the first 30–45 min. These results indicate that pretreatment with TRPA1 channel antagonist HC-030031 did significantly reduce the magnitude and shortened hyperalgesia, induced by SLIGRL. Some mirror image hyperalgesic effects which we observed on the contralateral paw can be explained as to be due to central sensitization at the spinal cord level. It is well known that peripheral and central sensitizations play important roles in the establishment of chronic pain and chronic itch. Chronic itch can be associated with spontaneous itch, hyperknesis (enhanced itch to a normally itchy stimulus), and alloknesis (itch elicited by an innocuous touch stimulus). Akiyama and Carstens [5] found in superficial dorsal horn neurons receiving afferent input from a dry skin-treated hindpaw a heightened spontaneous activity and enhanced responses to SLIGRL, but not histamine, compared to units recorded in control animals. We suppose that in our experiments such sensitization can activate neurons on the contralateral side of the spinal cord and induce some weaker hyperalgesia.

Concerning the involvement of TRPA1 channel in the heat hyperalgesia that could be due to sensitization of other nociceptive neurons that possess TRPA1, we can respond positively. It has been established that TRPA1 is the downstream target of both MrgprA3 and MrgprC11 in cultured sensory neurons and heterologous cells. TRPA1 is required for Mrgpr-mediated signaling, as sensory neurons from TRPA1-deficient mice exhibited markedly diminished responses to chloroquine and BAM8–22 [20]. Their behavioral studies also revealed a marked loss of itch-evoked behaviors in TRPA1-deficient animals in response to both of these pruritogens, thus indicating that TRPA1 is an essential component of the signaling pathways that promote histamine-independent itch [20]. In accordance with the results shown in the rodent model, patients with atopic dermatitis display increased expression of TRPA1 in both pruritic and non-pruritic skin. Considering that the TRPA1channel represents an important point of convergence for both pruriceptive and nociceptive signaling on C-fibers, its chronic upregulation could play an important role in hyperalgesia (the increased responses) to pruritic and mechano-nociceptive stimuli observed in patient suffering from atopic dermatitis [18].

Previously, we found similar effects of a significant attenuation of thermal and mechanical hyperalgesia induced by non-histaminergic pruritogens chloroquine and BAM8–22, injected into mice hindpaw, for the first 30–45 min. Pretreatment with HC-030031 did significantly reduce the magnitude of this hyperalgesia, as well as significantly shortened the time-course of hyperalgesia induced by chloroquine and BAM8–22 [25].

Recent studies have implicated TRPA1 channel as an intracellular regulator involved in itch [1,3,20,26,27,28,29,30]. The TRPA1 channel is thought to mediate chloroquine itching as the final common pathway and also to be present in sensory neurons in the skin, in mast cells, and in skin keratinocytes. The TRPA1 KO mice are insensitive to chloroquine- and BAM-mediated itch [20]. Thus, the TRPA1 is thought to mediate the transduction of itch neurotransmission in the skin to the nervous system. All these contribute to the peripheral neurotransmission of itch sensation to the DRG neurons [30].

The exciting adjustment to classical modes of GPCR signaling is expanding to a diverse collection of GPCRs and is in line with experiments showing abundant expression of the TRPA1 channel in nociception and itching [3,20]. There is elaborate “cross talk” among the diverse mediators and receptors involved in chloroquine-induced pruritus. Particularly, chloroquine binds to the MrgprA3/MrgprX1 receptors present in a small proportion (4–5%) of DRG neurons and skin. The MrgprA3 receptors are coupled to phospholipase C (PLC-*β*3) and a chloride channel to initiate skin itch action potentials in unmyelinated C-type nerve fibers. The Mrgpra3/X1 couples to TRPA1 for calcium influx into neuronal cells at non-cutaneous sites. Central chloroquine itch occurs via gastrin-related peptide (GRP) and its receptor (GRPR) in the dorsal spinothalamic tracts, as well as glutamic-mediated GRP projection to parabrachial nucleus [31].

The other series of experiments showed that TRPA1 was required for chloroquine- and BAM8–22-induced rises in intracellular calcium and action potentials in DRG neurons. No scratching was observed in TrpA1 null mice, and both MrgprA3 and MrgprC11 were shown to couple to TrpA1 in heterologous cells [20]. However, distinct mechanisms are involved in MrgprA3 and MrgprC11 signaling to activate TRPA1; Gβγ signaling is required for MrgprA3, while PLC signaling is required for MrgprC11 [20]. This in itself is another interesting modification of the classical GPCR models, in which the alpha subunits are the active messenger proteins [31]. At the basic science level, on the one hand, TRPA1 has been found to be evolutionarily conserved, with the involvement of TRPA1 orthologues in pain/itch-related behavior. On the other hand, from the point of view of the development of TRPA1 inhibitory/modulatory compounds for human clinical use, as of today, TRPA1 inhibitors have a shared feature in that they can all be improved in terms of better efficacy and fewer side effects [29,30]. Concerning the TRPA1 antagonists, there are a wide range of organic and inorganic chemicals characterized by heterogeneous chemotypes. The xanthine derivative known as HC-030031 can be considered the parent compound of most of the newer TRPA1 antagonists [32,33]. The expressing TRPA1 channel in HEK cells, HC-030031 was found to antagonize allyl isothiocyanate (AITC)- and formalin-evoked calcium influx [34]. Moreover, electrophysiology experiments by perforated-patch voltage-clamp recordings on TRPA1- expressing HEK293 cells confirmed that both inward and outward currents elicited by AITC or formalin were rapidly and reversibly blocked by this antagonist. IC50 values for TRPA1 blockade by HC-030031 or ruthenium red were similar to those observed in the Ca^2+^ imaging experiments [34]. The ability of HC-030031 to block TRPA1 activation was tested in a parallel fluorescent imaging plate reader (FLIPR) Ca^2+^-influx assay using HEK-293 cells stably expressing human TRPA1. This study has shown that HC-030031 dose-dependently blocked cinnamaldehyde- and AITC-induced Ca^2+^-influx [35].

The activity of HC-030031 was also evaluated on a number of rodent’s TRPA1 orthologues, confirming micromolar potency in different assays and against several agonists. The compound showed a higher selectivity for TRPA1 over almost 50 different targets (including enzymes, receptors, and transporters) involved in pain transmission [32,33]. Thus, HC-030031 is widely used as the prototypical TRPA1 antagonist and its employment as pharmacological tool largely contributed to the validation of the channel as drug target in multiple therapeutic areas [32,35], including pain and itch sensations.

## 5. Conclusions

Herein, we revealed that non-histaminergic pruritogen SLIGRL elicits thermal and mechanical hyperalgesia via the activation of TRPA1 channel. This hyperalgesia was attenuated by the TRPA1 channel antagonist HC-030031. Further studies are very important to get more evidence for the potential role of TRPA1 channel inhibitors as modulators of preclinical and/or clinical itch and pain conditions.

## Figures and Tables

**Figure 1 medsci-07-00062-f001:**
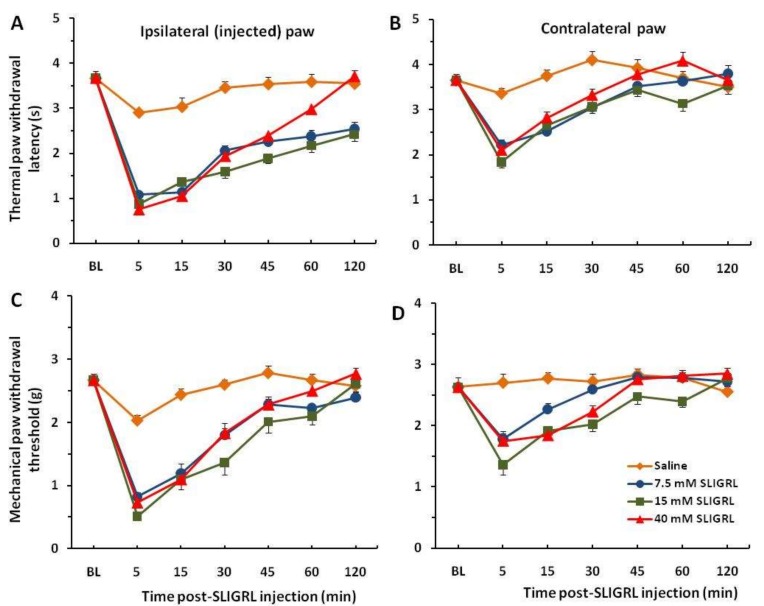
Dynamics of the thermal and mechanical withdrawals. Intra-plantar injection of Ser-Leu-Ile-Gly-Arg-Leu (SLIGRL) results in a significant decrease of the thermal paw latency (**A**) and mechanical paw threshold (**C**), i.e., develops hyperalgesia. There are weaker mirror hyperalgesic effects for the contralateral paw (**B**,**D**). BL, pre-injection baseline. Note, there are no dose-dependent effects of SLIGRL on hyperalgesia (**A**,**C**).

**Figure 2 medsci-07-00062-f002:**
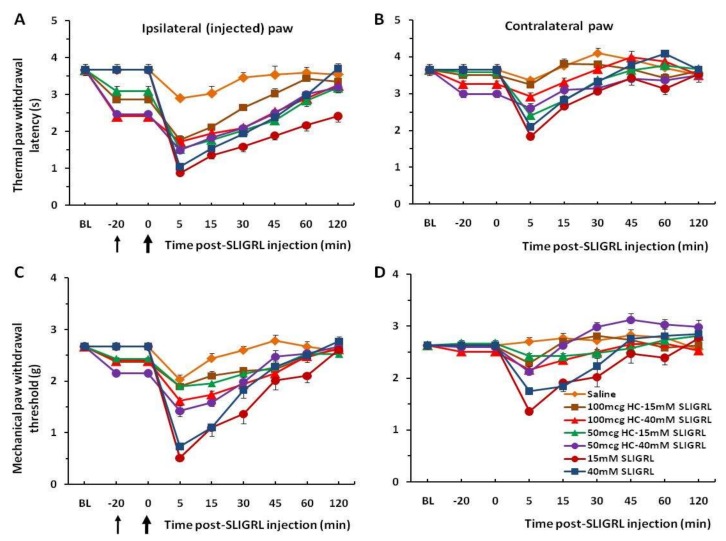
Dynamics of the thermal and mechanical withdrawals after HC-030031 pretreatment. Intra-plantar injection of SLIGRL results in a significant decrease of the thermal paw latency (**A**) and mechanical paw threshold (**C**), i.e., develops hyperalgesia. However, pretreatment with transient receptor potential ankyrin 1 (TRPA1) antagonist HC-030031 reduces these thermal and mechanical hyperalgesia, respectively. There are weaker mirror hyperalgesic effects for the contralateral paw (**B**,**D**). Note, there was a significant difference in the extent of inhibition between 50 mg and 100 mg HC-030031 for 15 mM SLIGRL at 30–60 min of the post-SLIGRL injection (*p* < 0.001), but not for 40 mM SLIGRL (*p* > 0.05) in the thermal test (**A**). The thin black arrow indicates the time of injection of HC-030031 and the bold arrow indicates the time of injection of SLIGRL. BL; pre-injection baseline.

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
