# Peer review of "TRPA1 Channel is Involved in SLIGRL-Evoked Thermal and Mechanical Hyperalgesia in Mice"

_medsci, 2019, doi:10.3390/medsci7040062_

Round 1
Reviewer 1 Report
This is a well done and timely study showing that a
pruritogen, SLIGRL, which excites MrgprC11, not only induces scratching
behavior but also produces mechanical and heat hyperalgesia. It was
also shown that TRPA1 contributes to this hyperalgesia as it was reduced
by a TRPA1 antagonist. The studies are well designed, the results are
presented clearly, and the discussion is appropriate. Thee are only a
few minor concerns/suggestions for improvement.
English could be improved in several areas of the text including the abstract.
The contralateral hyperalgesia is particulally interesting. The authors point out that this likely reflects central sensitization but it's not clear how this might occur. Would neurons in the contralateral spinal cord become sensitized? These interesting results could benefit from a little more discussion.
The development of hyperalgesia,
and that it's mediated by TRPA1 is also very interesting and novel. The
authors might want to discuss how his might occur. Do the same neurons
that mediate the scratching (itch) also mediate hyperalgesia? Or,
could the hyperalgesia be due to sensitization of other nociceptive
neurons that possess TRPA1?
Author Response
Response to Reviewer 1 Comments
Point 1. English could be improved in several areas of the text including the abstract.
Response 1: After completion of revision of the MS, I am going to send it to language editing by MDPI.
Point 2. The contralateral hyperalgesia is particularly interesting. The authors point out that this likely reflects central sensitization but it's not clear how this might occur. Would neurons in the contralateral spinal cord become sensitized? These interesting results could benefit from a little more discussion.
Response 2: Peripheral and central sensitizations play important roles in the establishment of chronic pain and chronic itch. Chronic itch can be associated with spontaneous itch, hyperknesis (enhanced itch to a normally itchy stimulus), and alloknesis (itch elicited by an innocuous touch stimulus). Akiyama and Carstens (2013) found in superficial dorsal horn neurons receiving afferent input from a dry skin-treated hindpaw heightened spontaneous activity and enhanced responses to SLIGRL, but not histamine, compared to units recorded in control animals [5]. We suppose that in our experiments such sensitization can activate neurons on the contralateral side of the spinal cord and induce some weaker hyperalgesia. I will add this paragraph to the manuscript.
Point 3. The development of hyperalgesia, and that it's mediated by TRPA1 is also very interesting and novel. The authors might want to discuss how his might occur. Do the same neurons that mediate the scratching (itch) also mediate hyperalgesia? Or, could the hyperalgesia be due to sensitization of other nociceptive neurons that possess TRPA1?
Response 3: Nowedays, it is difficult to say precisely, do the same neurons that mediate the scratching (itch) also mediate hyperalgesia. However, concerning hyperalgesia, could it be due to sensitization of other nociceptive neurons that possess TRPA1, we can respond positively. Wilson et al (2011) found that TRPA1 is the downstream target of both MrgprA3 and MrgprC11 in cultured sensory neurons and heterologous cells. TRPA1 is required for Mrgpr-mediated signaling, as sensory neurons from TRPA1-deficient mice exhibited markedly diminished responses to chloroquine and BAM8–22. Their behavioral studies also revealed a marked loss of itch-evoked behaviors in TRPA1-deficient animals in response to both of these pruritogens, thus, indicating that TRPA1 is an essential component of the signaling pathways that promote histamine-independent itch [20]. In accordance with the results showed in the rodent model, patients with atopic dermatitis display increased expression of TRPA1 in both pruritic and non-pruritic skin. Considering that the TRPA1channel represents an important point of convergence for both pruriceptive and nociceptive signaling on C-fibers, its chronic upregulation could play an important role in hyperalgesia (the increased responses) to pruritic and mechano-nociceptive stimuli observed in patient suffering from atopic dermatitis [18]. I will add this paragraph to the manuscript.

Reviewer 2 Report
@font-face { font-family: "Times"; }@font-face { font-family: "MS 明朝"; }@font-face { font-family: "MS 明朝"; }@font-face { font-family: "@MS 明朝"; }@font-face { font-family: "MS P明朝"; }@font-face { font-family: "@MS P明朝"; }@font-face { font-family: "平成明朝"; }@font-face { font-family: "@平成明朝"; }p.MsoNormal, li.MsoNormal, div.MsoNormal { margin: 0mm 0mm 0; text-align: justify; font-size: 16px; font-family: "MS 明朝"; }.MsoChpDefault { font-size: 13px; font-family: "MS 明朝"; }div.WordSection1 { }ol { margin-bottom: 0mm; }ul { margin-bottom: 0mm; }
Merab Tsagareli et al. examine whether thermal and mechanical hyperalgesia produced by a non-histaminergic pruritogen SLIGRL in mice are mediated by TRPA1 channels by using the behavioral technique. The thermal and mechanical hyperalgesia were measured from thermal paw withdrawal latencies and mechanical thresholds, respectively. As a result, they found out that SLIGRL produces thermal and mechanical hyperalgesia (Fig. 1) and this action is reversed by a TRPA1 antagonist HC-030031 (Fig. 2). There are a lot of points that need improvements in scientific writing and data presentation, as follows:
Major comments:
1. Lines 164-168: the authors mention that the same results as those of SLIGRL in the present study were obtained by using chloroquine and BAM8-22 [25]. If so, this fact should be clearly mentioned in Introduction.
2. Key words should contain TRPA1 and SLIGRL.
3. Line 145: the authors should mention why these concentrations of HC-030031 were used. Was there a difference in the extent of inhibition between 50 mg and 100 mg of HC-030031 in Fig. 2? Please reply to this question.
4. Figures 1 and 2 are confusing. It may be better to give “Ipsilateral (injected) paw” and “Contralateral paw” in C and D, respectively, in each of Figs. 1 and 2. In the legend of Fig. 1, there is no explanation about (B), (D) and BL. There are data about 7.5 mM SLIGRL in Fig. 1 but not Fig. 2. 38 mM SLIGRL is used in Fig. 1 while 40 mM SLIGRL is used. It is necessary for these facts to be explained. Please correct this point.
5. The concentrations of SLIGRL used in Fig. 2 are different from those mentioned in 2.2 Chemical injections. Please amend this point.
6. Line 160: the authors did not perform a quantitative analysis of the time course. This sentence should be revised.
7. The authors used several concentrations of SLIGRL in Figs. 1 and 2. There is no description about a concentration dependency for this effect. Please amend this point.
8. In the last paragraph on page 5 and the first paragraph on page 6, the authors mention the actions of chloroquine and BAM8-22 but not SLIGRL. Please amend this point.
9. The authors should check this manuscript very carefully. There are many English and scientific mistakes.
Author Response
Response to Reviewer 2 Comments
Point 1: Lines 164-168: the authors mention that the same results as those of SLIGRL in the present study were obtained by using chloroquine and BAM8-22 [25]. If so, this fact should be clearly mentioned in Introduction.
Response 1: Thanks, yes, I have added the sentence in the manuscript: “Recently, we have found that pretreatment with TRPA1 channel antagonist HC-030031 did significantly reduce hyperalgesia evoked by chloroquine and BAM8-22, injected into mice hindpaw”.
Point 2. Key words should contain TRPA1 and SLIGRL.
Response 2: As these words are in the Title of the MS, I did not put them in Keywords, but now I will add them.
Point 3. Line 145: the authors should mention why these concentrations of HC-030031 were used. Was there a difference in the extent of inhibition between 50 mg and 100 mg of HC-030031 in Fig. 2? Please reply to this question.
Response 3: There was a significant difference in the extent of inhibition between 50 mg and 100 mg HC-030031 for 15 mM SLIGRL at 30-60 min of the post-SLIGRL injection (P < 0.001), but not for 40 mM SLIGRL (p > 0.05) in the thermal test (Fig. 1A). I will add this sentence in the manuscript and in the legend of Fig. 2.
Point 4. Figures 1 and 2 are confusing. It may be better to give “Ipsilateral (injected) paw” and “Contralateral paw” in C and D, respectively, in each of Figs. 1 and 2. In the legend of Fig. 1, there is no explanation about (B), (D) and BL. There are data about 7.5 mM SLIGRL in Fig. 1 but not Fig. 2. 38 mM SLIGRL is used in Fig. 1 while 40 mM SLIGRL is used. It is necessary for these facts to be explained. Please correct this point.
Response 4: Thanks, I have corrected the highest concentration of SLIGRL into 40 mM SLIGRL and added in the legend of Fig. 1, about (B), (D) and BL.
Point 5. The concentrations of SLIGRL used in Fig. 2 are different from those mentioned in 2.2 Chemical injections. Please amend this point.
Response 5: Thanks, yes, I have corrected it in the Method section too.
Point 6. Line 160: the authors did not perform a quantitative analysis of the time course. This sentence should be revised.
Response 6: Thanks, I corrected this sentence: “These results indicate that pretreatment with TRPA1 channel antagonist HC-030031 did significantly reduce the magnitude and of shortened hyperalgesia, induced by SLIGRL”.
Point 7. The authors used several concentrations of SLIGRL in Figs. 1 and 2. There is no description about a concentration dependency for this effect. Please amend this point.
Response 7: Yes, you are right, on these data, the effects of hyperalgesia are not dose-depending. In the next experiments the concentration of SLIGRL should be reduced.
Point 8. In the last paragraph on page 5 and the first paragraph on page 6, the authors mention the actions of chloroquine and BAM8-22 but not SLIGRL. Please amend this point.
Response 8: This is literature data showing same effects of chloroquine and BAM8-22 on TRPA1 like as SLIGRLE does, and I kindly suppose to leave this paragraph in the manuscript.
Point 9. The authors should check this manuscript very carefully. There are many English and scientific mistakes.
Response 9: After completion of revision of the MS, I am going to send it to language editing by MDPI.

Reviewer 3 Report
Tsagareli et al. explored an involvement of TRPA1 in the process of histamine independent itch MrgprC11 mediated itch by its agonists SLIGRL. The authors showed SLIGRL injection to paw induced thermal and mechanical paw withdrawal behavior under observatory time of 120 minutes. And the behavior evoked by SLIGRL was partially suppressed by TRPA1 antagonist HC 030031.
The main criticism is whether the behavior caused by SLIGRL could assume itch or not, in these sets of experiments. von Frey test and thermal paw withdrawal test are the mechanical or thermal hyperalgesia as the authors mentioned. It is not clear the authors use pain and itch in this context.
The authors stated significant difference between groups they used, however, it was not indicated in the figures between which groups showed difference.
Author Response
Response to Reviewer 3 Comments
Point 1. The main criticism is whether the behavior caused by SLIGRL could assume itch or not, in these sets of experiments. von Frey test and thermal paw withdrawal test are the mechanical or thermal hyperalgesia as the authors mentioned. It is not clear the authors use pain and itch in this context.
Response 1: The main idea of this short report was whether a pruritogen SLIGRL could induce hyperalgesia? It is well known that SLIGRL produces MrgprC11 mediated itch in mice and we are interested this process is whether accompanied by hyperalgesia or not. We continuous this research to get more data for full paper.
Point 2. The authors stated significant difference between groups they used; however, it was not indicated in the figures between which groups showed difference.
Response 2: In order to avoid some complications in the understanding of graphs, we did not indicate differences between the groups. We present this information in the text of MS and in the legends of pictures.

Reviewer 4 Report
The manuscript by Tsagareli and collaborators entitled "SLIGRL evoked-itch is accompanied by thermal and mechanical hyperalgesia in mice: the role of TRPA1 channel” reports simple data regarding the involvement of the TRPA1 channel in mechanical and thermal hyperalgesia after intraplantar SLIGRL
Data are very poor, they could be interesting but, there are many issues that need to be addressed by the authors.
1. The title “SLIGRL evoked-itch is accompanied by thermal and mechanical hyperalgesia in mice: the role of TRPA1 channel” is not suitable for the data reported, since author did not report the SLIGRL evoked-itch thus I think that it should be changed “TRPA1 channel is involved in SLIGRL evoked thermal and mechanical hyperalgesia in mice”.
2. Authors should discuss the involvement of TRPA1 channel in the heat hyperalgesia. Does it refer to general sensitization? Please discuss it.
3. To better study the role of TRPA1 channel authors need to test SLIGRL in TRPA1 wt and KO mice.
4. Abstract Line 21 HC030031 should be changed in HC-030031 as reported throughout the manuscript
5. Line 61 Please spelled out name of single family channel
In general, mammalian TRP channels comprise six related protein subfamilies (TRPC, TRPV, TRPM, TRPA, TRPML, TRPP).
6. Line 71 “most pruritic stimuli” “most pruritogenic stimuli”
7. Line 77 BAM should be changed in BAM8–22 as reported throughout the manuscript
8. Line 93 “The experiments were performed on male mice < 50 grams”. Please specify mice strains
9. Line 100 “The peptide SLIGRL (7.5, 15, and 38 mM /1μL), and TRPA1 antagonist HC030031 (50μg/30μL; 100 100μg/30μL) were purchased from Sigma-Aldrich Chemicals, Co., (St. Louis, MO, USA).” Please rephrase as:
SLIGRL, and the TRPA1 antagonist HC030031 were purchased from Sigma-Aldrich Chemicals, Co., (St. Louis, MO, USA). SLIGRL was dissolved in xxxx to obtain 7.5, 15, and 38 mM /1μL, final dose, and HC030031 was dissolved in to obtain final dose of 50μg and 100μg/30μL
10. Line 102 please change intradermally with intraplantar as reported throughout the manuscript
11. Line 102 a 30 g needle is 30G needle
12. Line 121 and 127 mash is mesh
13. Please insert statistical symbols in the graphs
14. Fig 1 there is not a clear a dose response either in mechanical and thermal hyperalgesia. how can be explained?
15. Fig 1 and Fig 2 legends, please insert statical
16. Based on the data reported, the local injection of SLIGRL induced a mechanical and heat hyperlagia both in ipsilateral and contralateral paw. I do not understand if in the contralateral paw the reduction in the mechanical and thermal threshold is statistical different from the saline injected paw (control). But, more importantly if HC030031 ipsilateral injected is able to reduce mechanical and thermal hyperalgesia in the contralateral paw. Please specify and discuss this issue or delete the contralateral data. They are misleading
17. Line 171 path is pathway
18. Line 172 “TRPA1 deficient or KO mice do not react to chloroquine with itch [20]”. Please verify.
19. Line 183 pretein is protein
20. Line 193 TrpA1 is TRPA1 as reported throughout the manuscript
Author Response
Response to Reviewer 4 Comments
Point 1. The title “SLIGRL evoked-itch is accompanied by thermal and mechanical hyperalgesia in mice: the role of TRPA1 channel” is not suitable for the data reported, since author did not report the SLIGRL evoked-itch thus I think that it should be changed “TRPA1 channel is involved in SLIGRL evoked thermal and mechanical hyperalgesia in mice”.
Response 1: Thanks, yes, I agree with you and I will change the title as you indicated it.
Point 2. Authors should discuss the involvement of TRPA1 channel in the heat hyperalgesia. Does it refer to general sensitization? Please discuss it.
Response 2: Concerning the involvement of TRPA1 channel in the heat hyperalgesia, could it be due to sensitization of other nociceptive neurons that possess TRPA1, we can respond positively. Wilson et al (2011) found that TRPA1 is the downstream target of both MrgprA3 and MrgprC11 in cultured sensory neurons and heterologous cells. TRPA1 is required for Mrgpr-mediated signaling, as sensory neurons from TRPA1-deficient mice exhibited markedly diminished responses to chloroquine and BAM8–22. Their behavioral studies also revealed a marked loss of itch-evoked behaviors in TRPA1-deficient animals in response to both of these pruritogens, thus, indicating that TRPA1 is an essential component of the signaling pathways that promote histamine-independent itch [20]. In accordance with the results showed in the rodent model, patients with atopic dermatitis display increased expression of TRPA1 in both pruritic and non-pruritic skin. Considering that the TRPA1channel represents an important point of convergence for both pruriceptive and nociceptive signaling on C-fibers, its chronic upregulation could play an important role in hyperalgesia to pruritic and nociceptive stimuli observed in patient suffering from atopic dermatitis [18]. I will add this paragraph to the manuscript.
Point 3. To better study the role of TRPA1 channel authors need to test SLIGRL in TRPA1 wt and KO mice.
Reponse 3: We continue this research on this direction. Thanks.
Point 4. Abstract Line 21 HC030031 should be changed in HC-030031 as reported throughout the manuscript
Response 4. Thanks, I have corrected it.
Point 5. Line 61 Please spelled out name of single family channel. In general, mammalian TRP channels comprise six related protein subfamilies (TRPC, TRPV, TRPM, TRPA, TRPML, TRPP).
Response 5: Thanks, I corrected it as: “TRP-Canonical (TRPC), TRP-Vanilloid (TRPV), TRP-Melastatin (TRPM), TRP-Ankyrin (TRPA), TRP-Mucolipin (TRPML) and TRP-Polycystic).
Point 6. Line 71 “most pruritic stimuli” “most pruritogenic stimuli”
Response 6: Thanks, I have corrected it.
Point 7. Line 77 BAM should be changed in BAM8–22 as reported throughout the manuscript
Response 7: Thanks, I have corrected it.
Point 8. Line 93 “The experiments were performed on male mice < 50 grams”. Please specify mice strains
Response 8: Mice are wild type and I put it in the MS.
Point 9. Line 100 “The peptide SLIGRL (7.5, 15, and 38 mM /1μL), and TRPA1 antagonist HC030031 (50μg/30μL; 100 100μg/30μL) were purchased from Sigma-Aldrich Chemicals, Co., (St. Louis, MO, USA).” Please rephrase as:
SLIGRL, and the TRPA1 antagonist HC030031 were purchased from Sigma-Aldrich Chemicals, Co., (St. Louis, MO, USA). SLIGRL was dissolved in xxxx to obtain 7.5, 15, and 38 mM /1μL, final dose, and HC030031 was dissolved in to obtain final dose of 50μg and 100μg/30μL
Response 9: Thanks, I have rephrased this sentence: “The peptide SLIGRL and the TRPA1 antagonist HC-030031 were purchased from Sigma-Aldrich Chemicals, Co., (St. Louis, MO, USA). SLIGRL was dissolved in saline to obtain 7.5, 15, and 40 mM /1μL), final dose, and HC-030031 was dissolved in 30μL 1% DMSO and saline to obtain final dose 50μg and 100μg/30μL.”
Point 10. Line 102 please change intradermally with intraplantar as reported throughout the manuscript
Response 10: Thanks, I have corrected it.
Point 11. Line 102 a 30 g needle is 30G needle
Response 11: Thanks, I have corrected it.
Point 12. Line 121 and 127 mash is mesh
Response 12: Thanks, I have corrected it.
Point 13. Please insert statistical symbols in the graphs
Response 13: In order to avoid some complications in the understanding of graphs, we did not indicate differences between the groups. We present this information in the text of MS and in the legends of pictures.
Point 14. Fig 1 there is not a clear a dose response either in mechanical and thermal hyperalgesia. how can be explained?
Response 14: Yes, on these data, the effects of hyperalgesia are not dose-depending. In the next experiments the concentration of SLIGRL should be reduced.
Point 15. Fig 1 and Fig 2 legends, please insert statical
Response 15: Thanks. I have inserted statistical data in the legends.
Point 16. Based on the data reported, the local injection of SLIGRL induced a mechanical and heat hyperalgesia both in ipsilateral and contralateral paw. I do not understand if in the contralateral paw the reduction in the mechanical and thermal threshold is statistical different from the saline injected paw (control). But, more importantly if HC030031 ipsilateral injected is able to reduce mechanical and thermal hyperalgesia in the contralateral paw. Please specify and discuss this issue or delete the contralateral data. They are misleading
Response 16: Peripheral and central sensitizations play important roles in the establishment of chronic pain and chronic itch. Chronic itch can be associated with spontaneous itch, hyperknesis (enhanced itch to a normally itchy stimulus), and alloknesis (itch elicited by an innocuous touch stimulus). Akiyama and Carstens (2013) found in superficial dorsal horn neurons receiving afferent input from a dry skin-treated hindpaw heightened spontaneous activity and enhanced responses to SLIGRL, but not histamine, compared to units recorded in control animals [5]. We suppose that in our experiments such sensitization can activate neurons on the contralateral side of the spinal cord and induce some weaker hyperalgesia. I will add this paragraph into the manuscript.
Point 17. Line 171 path is pathway
Response 17: Thanks, I have corrected it.
Point 18. Line 172 “TRPA1 deficient or KO mice do not react to chloroquine with itch [20]”. Please verify.
Response 18: Thanks, I have corrected this phrase into: “TRPA1 KO mice are insensitive to chloroquine- and BAM-mediated itch [20].
Point 19. Line 183 pretein is protein
Response 19: Thanks, I have corrected it.
Point 20. Line 193 TrpA1 is TRPA1 as reported throughout the manuscript
Response 20: Thanks, I have corrected it.

Round 2
Reviewer 2 Report
This is a revised manuscript written by Merab G. Tsagareli who investigated whether thermal or mechanical hyperalgesia produced by a non-histaminergic pruritogen SLIGRL (PAR-2 agonist) in mice is mediated by TRPA1 channels by using a TRPA1 antagonist HC-030031 and the behavioral technique. This appears to be not fully amended compared to the original manuscript. Data shown in this manuscript were probably obtained together with those published in [25], although [25]’s first author was Ivliance Nozadze (the second author of the present manuscript). This is clear from a manuscript titled as “Non-Histaminergic Itch is Accompanied by Thermal and Mechanical Hyperalgesia in Mice: The Role of TRPA1 Channel” (where the involvement of chloroquine, BAM8-22 and SLIGRL in itch was given), submitted previously to Medical Sciences. Although data shown in the present manuscript and [25] are obtained by the same technique, this fact has not been described in the Materials and Methods section in this manuscript. There is only a few data shown in two figures in this manuscript. However, the Discussion section is unusually lengthy, and has almost not mentioned an involvement of SLIGRL in itch, whereas an involvement of BAM8-22 and SLIGRL is largely mentioned in lines 245-261. Alternatively, the authors have described a detail of HC-030031 in lines 267-318, many of which contents are not directly related to this manuscript. Here, a phrase, “inward ad outward currents elicited by AITC” in line 272 could not be understood without explanation. Furthermore, there are many problems in terms of English and scientific writing. For example,
Line 11: not “select” but “selective”?
Line 14: not “MrgprC11respond” but “MrgprC11 responds”?
Lines 26, 116, 123, 171, 194, 209, 235, 322: HC-030031 is explained many times as a TRPA1 antagonist. Only one explanation of HC-030031 is OK, because HC-030031 is well-known and widely used as a TRPA1 antagonist.
Line 66: TRPP should not be deleted.
Line 82: not “ganglia” but “ganglion”?
Line 152: it is not clear from the Result section whether Kolmogorov-Smirnov test was used in this study.
The legends of Figs. 1 and 2: there is not title in these legends. The first sentences in the two legends were quite the same.
Line 195: “of” should be deleted.
Line 196: not “what” but “which”?
Lines 216-224: there is no reference here.
Line 227: not “showed” but “shown”?
Line 245: it is not necessary to define GPCR here (see line 40).
Lines 249-250: not “dorsal root ganglion” but “DRG”?
Author Response
Thank you very much for your kind notes and comments which much improved our manuscripts.
Response 1. I put the sentence: “Experimental procedures were performed as described previously [25]” in the method sections, line 102. Thanks.
Response 2. I have corrected the sentences:
“Moreover, electrophysiology experiments by perforated-patch voltage–clamp recordings on TRPA1- expressing HEK293 cells, confirmed that both inward and outward currents elicited by AITC or formalin were rapidly and reversibly blocked by this antagonist. IC50 values for TRPA1 blockade by HC-030031 or ruthenium red were similar to those observed in the Ca2+ imaging experiments [34].” Lines 243-248.
Line 11: not “select” but “selective”?
Yes, I have corrected into “selective”, thanks.
Line 14: not “MrgprC11respond” but “MrgprC11 responds”?
Yes, I have corrected it, “MrgprC11 responds”, thanks.
Lines 26, 116, 123, 171, 194, 209, 235, 322: HC-030031 is explained many times as a TRPA1 antagonist. Only one explanation of HC-030031 is OK, because HC-030031 is well-known and widely used as a TRPA1 antagonist.
Yes, I have corrected them, just left this phrase in sections – Absract, Introduction, and Methods. Thanks.
Line 66: TRPP should not be deleted.
Yes, I have added it. Thanks.
Line 82: not “ganglia” but “ganglion”?
Yes, I have corrected it, “ganglion”, thanks.
Line 152: it is not clear from the Result section whether Kolmogorov-Smirnov test was used in this study.
Yes, I have removed this sentence. Thanks.
The legends of Figs. 1 and 2: there is not title in these legends. The first sentences in the two legends were quite the same.
I have added the titles for both Figures. Thanks
Line 195: “of” should be deleted.
Yes, I deleted it, thanks.
Line 196: not “what” but “which”?
Yes, I corrected it, thanks.
Lines 216-224: there is no reference here.
Yes, I put reference there, thanks.
Line 227: not “showed” but “shown”?
Yes, I have corrected it.
Line 245: it is not necessary to define GPCR here (see line 40).
Yes, I have corrected it, remove its definition.
Lines 249-250: not “dorsal root ganglion” but “DRG”?
I have corrected it, into DRG, thanks.
After completion of revision of the MS, I am going to send it to language editing by MDPI.
Reviewer 3 Report
The aim of this study is to demonstrate the involvement of SLIGRL peptides, agonist ofMas-related G-protein-coupled receptors, to thermal or mechanical hyperalgesia of mice hindpaw, and it is modulated by TRPA1 antagonist.
My main concern is that it is not clearly demonstrated statistically the significant differences between which conditions such as different time points and different concentrations. In discussion, the authors should discuss the mechanism of TRPA1 and TRPV1 involvement to itch behavior and mechanical or thermal sensitivity.
Author Response
Thank you very much for your all, previous and current kind notes and comments which much improve our manuscript.
Response : As TRPV1 is involved in histaminergic itch, I did not discusses this issue here, and also other reviewers said, that Discussion section is lengthy, thanks.
This manuscript is a resubmission of an earlier submission. The following is a list of the peer review reports and author responses from that submission.
Round 1
Reviewer 1 Report
@font-face { font-family: "Times"; }@font-face { font-family: "MS 明朝"; }@font-face { font-family: "MS 明朝"; }@font-face { font-family: "@MS 明朝"; }@font-face { font-family: "平成明朝"; }@font-face { font-family: "MS P明朝"; }@font-face { font-family: "@MS P明朝"; }@font-face { font-family: "@平成明朝"; }p.MsoNormal, li.MsoNormal, div.MsoNormal { margin: 0mm 0mm 0; text-align: justify; font-size: 16px; font-family: "MS 明朝"; }.MsoChpDefault { font-size: 13px; font-family: "MS 明朝"; }div.WordSection1 { }ol { margin-bottom: 0mm; }ul { margin-bottom: 0mm; }
Merab Tsagareli et al. examine whether thermal and mechanical hyperalgesia produced by non-histaminergic pruritogens (chloroquine, BAM8-22 and SLIGRL) in mice are mediated by TRPA1 channels by using the behavioral technique. The thermal and mechanical hyperalgesia were measured from thermal paw withdrawal latencies and mechanical thresholds, respectively. As a result, they found out that a TRPA1 antagonist HC-030031 inhibits thermal and hyperalgesia produced by chloroquine (Fig. 1), BAM8-22 (Fig. 2) and SLIGRL (Fig. 3). There are many points that may serve to amend this manuscript, as follows:
Major comments:
1. Line 152: the authors should mention why these concentrations of HC-030031 were used. Was there a difference in the extent of inhibition between 50 mg and 100 mg of HC-030031 in Figs. 1-3? Please reply to this question.
2. All of the figures are somewhat confusing. It would be better to give “Ipsilateral (injected) paw” and “Contralateral paw” in C and D, respectively, in each of Figs. 1-3.
3. The concentrations of BAM8-22 and SLIGRL used in Figs. 2 and 3 are different from those mentioned in 2.2 Chemical injections. Which values are right? Please reply to this question.
4. The authors used several concentrations of chloroquine, BAM8-22 and SLIGRL in Figs. 1-3. There is no description about a concentration dependency for their effects. Please amend this point.
5. Discussion section except for the second paragraph from the below on page 6 is not directly related to the experimental results obtained by the authors. This section should be shortened.
6. The authors should check the manuscript very carefully. There are many English and scientific mistakes, only a part of which is pointed out in minor comments.
Minor comments:
1. Line 12: is “…bind select pruritogens…” OK? Please correct this sentence.
2. Line 14: not “are” but “is”; not “respond” but “responds”.
3. Line 150: is “…significant compared…” OK? Please correct this sentence.
4. Line 201: is “…due as to be to…” OK? Please correct this sentence.
5. Line 207: is “itch neurotransmission.” OK? Please correct this sentence.
6. Line 208: is “…peripheral neurotransmission…” OK? Please correct this sentence.
7. Line 209: it is not necessary to repeatedly define GPCR (see line 38). Please amend this point.
8. Lines 213-214: please use here “DRG” (see line 75).
9. Line 215: is “…skin itch action potentials…” OK? Please correct this sentence.
10. Line 221: the second “TrpA1” should be “TRPA1”. Please correct this point.
11. Lines 233-234: is “The expressing TRPA1 channel…” OK? Please correct this sentence.
12. Line 236: what is “inward and outward currents elicited by AITC or formalin”? This sentence should be revised. Reference [33] appears to show only an inward current. Please amend this point.
13. Line 239: although the authors strengthen a dose-dependency of HC-030031, they themselves do not clearly show this dose-dependency. Please amend this point.
14. Lines 250-251: the involvement of TRPA1 channels in this study was judged from an inhibition by HC-030031. Therefore, this sentence should be amended.